# Mechanical Properties and Weibull Scaling Laws of Unknown Spider Silks

**DOI:** 10.3390/molecules25122938

**Published:** 2020-06-26

**Authors:** Gabriele Greco, Nicola M. Pugno

**Affiliations:** 1Laboratory of Bio-inspired, Bionic, Nano, Meta Materials & Mechanics, Department of Civil, Environmental and Mechanical Engineering, University of Trento, Via Mesiano, 77, 38123 Trento, Italy; gabriele.greco-2@unitn.it; 2Queen Mary University of London, Mile End Rd, London E1 4NS, UK

**Keywords:** statistics, Weibull statistics, effect size, strain rates

## Abstract

Spider silks present extraordinary mechanical properties, which have attracted the attention of material scientists in recent decades. In particular, the strength and the toughness of these protein-based materials outperform the ones of many man-made fibers. Unfortunately, despite the huge interest, there is an absence of statistical investigation on the mechanical properties of spider silks and their related size effects due to the length of the fibers. Moreover, several spider silks have never been mechanically tested. Accordingly, in this work, we measured the mechanical properties and computed the Weibull parameters for different spider silks, some of them unknown in the literature. We also measured the mechanical properties at different strain rates for the dragline of the species *Cupiennius salei*. For the same species, we measured the strength and Weibull parameters at different fiber lengths. In this way, we obtained the spider silk scaling laws directly and according to Weibull’s prediction. Both length and strain rates affect the mechanical properties of spider silk, as rationalized by Weibull’s statistics.

## 1. Introduction

Spider silks are biocompatible materials that present extraordinary mechanical properties in comparison to most of the natural and manmade fibers [1,2,3,4,5,6]. For this reason, they have been deeply investigated as potential materials in biomedical technologies, e.g., tissue engineering [7,8,9,10]. Spider silks are produced by different glands with different aims [11,12]. Because of this, they have evolved to have different mechanical properties, up to the species and individual gland that spins the silk [13,14,15,16]. Considering that more than 48,000 species of spiders exist, it is interesting to investigate, across different species and glands, the mechanical properties of the silks [17,18] (Table 1). Usually, these are Young’s modulus, toughness modulus, ultimate strain and fracture strength, which are obtained by means of tensile tests, i.e., stress-strain curves [19].

Exploring the mechanical properties of spider silks is difficult, because of their dependence on the testing parameters and their natural variability. Spinning forces [20], intrinsic variability [21], environmental humidity [22], temperature [23], possible contact with strong polar solvents (e.g., water) [24] and strain rates [19,25] are examples of factors to take into consideration, and which have been investigated [26].

In structural engineering, the role of defects has a huge importance in predicting the strength of materials [27,28]. Spider silks are self-assembly materials that, after their formation, could be affected by defects. Since the presence of these is strictly correlated with the size of the sample, thin fibers such as spider silk are expected to be intrinsically strong, but the length of the silk sample may affect its mechanical properties [28,29]. Thus, it is important to investigate the silk’s mechanical properties at different length scales. In this context, Weibull statistics have been widely used in past and recent years, to describe the statistical distribution of the ultimate strength of many natural or artificial materials (not necessarily brittles), including the silk of the spider *Argiope trifasciata* [21,30]. The importance of this statistic is mainly due to its versatility in describing different phenomena, and the possibility of using it to obtain scaling laws for the analyzed material. The Weibull’s distribution [31] is defined with its cumulative density function [thus *F*(*x*) represents the probability of the fracture strength being equal to or less than *x*], as:(1)$$F\left(x\right)=1-e^{-{\left(\frac{l}{l_{0}}\right)}^{d}{\left(\frac{x}{x_{0}}\right)}^{c}}=1-e^{-{\left(\frac{x}{b}\right)}^{c}}$$
where, in our case, *x* is the fracture strength of the material, *c* is the shape parameter, *x*_0_ is the scale parameter associated with the strength, *l* is the length of the sample, *l*_0_ is a characteristic length and *d* is the topological dimension. The dimension *d* expresses how the energy is dissipated during fracture, namely if it is dissipated in a volume (*d* = 3), over an area (*d* = 2) or along the length (*d* = 1). Here, it can be considered as Weibull “fractal dimension”, 0 < *d* < 3, which we expect to be close to 1 due to the mainly unidimensional nature of the fiber. Equally, it represents the dimension of the dispersion of defects in the material. 

Fixing the argument of the exponential to a constant (*k*) would result in the prediction of the scaling law, for the strength *x*, as proportional ($-k^{\frac{1}{c}}$) to *b*, with
(2)$$x\propto b,\;b=x_{0}{\left(\frac{l}{l_{0}}\right)}^{-\frac{d}{c}}$$

In this work, the mechanical properties and Weibull parameters of the silk of nine species, some of them still absent in the literature, were derived through tensile tests. Moreover, for the species *Cupiennius salei*, we studied scale and shape parameters at various lengths of the samples, and the effect of strain rates on the mechanical properties. We found that both length and strain rates affect the mechanical properties of spider silk, and Weibull’s statistics can help in their understanding. 

## 2. Results

The mechanical properties and Weibull parameters of the silk of nine species of spiders were obtained by means of their stress-strain curves (Appendix A, Table 1), which showed the typical nonlinear constitutive law of the silk [11]. Among them, the mechanical properties of the species *Ancylometes bogotensis*, *Ceratogyrus marshalli*, *Linothele fallax* and *Phoneutria fera* were absent in the literature. The other species analyzed in this work presented mechanical properties comparable with those in the literature [32,33,34]. Furthermore, for the first time we report the Weibull parameters for these nine species. The scale parameter is related to the strength of the silk, and the ones obtained were comparable to the mean values of the analyzed silks’ strength. The shape parameters, on the other hand, can be separated into two groups. Shape parameters lower than 2 were found for the species *Ceratogyrus marshalli*, *Grammostola rosea* and *Linothele fallax*. These belong to the infraorder of Mygalomorphae, which does not have major ampullate silk (the main component of the dragline). Shape parameters higher than 2 were found for all the other species, which belong to the infraorder of Araneomorphae and which have major ampullate silk. 

Figure 1 and Table 2 show the results related to the dragline silk of *Cupiennius salei*, tested at different strain rates (0.08 mm/s, 0.10 mm/s, 0.11 mm/s, 0.15 mm/s and 0.17 mm/s). The strain at break, displayed in Figure 1a (*p*-values and *d_c_* coefficient in Appendix A), was significantly higher at the strain rate 0.11 mm/s than that at 0.08 mm/s, 0.10 mm/s and 0.17 mm/s (with a medium effect size, ES). Respectively, the obtained values were 0.21 ± 0.15, 0.21 ± 0.12, 0.29 ± 0.15, 0.23 ± 0.17 and 0.20 ± 0.11.

For the strength, displayed in Figure 1b (*p*-values and *d_c_* coefficient in Appendix A), we observed significantly higher values at the strain rate 0.15 mm/s with respect to all the others (with a large ES). Respectively, the obtained values were 288 ± 241 MPa, 289 ± 218 MPa, 253 ± 217 MPa, 510 ± 311 MPa and 259 ± 168 MPa.

Similarly, for the Young’s modulus (Figure 1c, *p*-values and *d_c_* coefficient in Appendix A), we observed a significantly higher value for the strain rate of 0.10 mm/s, with respect to the 0.08 mm/s strain rate (with a small ES). The strain rate of 0.15 mm/s presented the highest Young’s modulus by a significant degree, compared to 0.08 mm/s (very large ES), 0.10 mm/s (medium ES), 0.11 mm/s (very large ES) and 0.17 mm/s (medium ES). The 0.11 mm/s strain rate gave lower values, compared to 0.08 mm/s (with a large ES), 0.10 mm/s (with a large ES) and 0.17 mm/s (with a large ES). Finally, the strain rate of 0.17 mm/s gave lower Young’s modulus values with respect to the strain rate of 0.10 mm/s (with a very small ES). Respectively, the obtained values were 6.5 ± 3.9 GPa, 8.8 ± 6.5 GPa, 3.5 ± 2.8 GPa, 13.5 ± 7.0 GPa and 8.6 ± 5.4 GPa.

The toughness modulus (Figure 1d, *p*-values and *d_c_* coefficient in Appendix A) measured at 0.15 mm/s was significantly higher compared to all the others (with a medium ES). Respectively, the obtained values were 36 ± 41 MJ/m^3^, 45 ± 46 MJ/m^3^, 37 ± 35 MJ/m^3^, 76 ± 63 MJ/m^3^ and 37 ± 36 MJ/m^3^. 

Figure 2 and Table 3 show the results relative to the analyses of the mechanical properties and Weibull parameters at different lengths (0.55 cm, 0.75 cm, 1.0 cm, 1.25 cm and 1.5 cm). We did not observe any statistically significant differences in the strain at break, Young’s modulus or toughness modulus for the fibers tested at different lengths. For the strain at break, we measured respectively 0.23 ± 0.11, 0.20 ± 0.15, 0.25 ± 0.14, 0.22 ± 0.10 and 0.21 ± 0.17. For the Young’s modulus, we obtained respectively 8.0 ± 4.1 GPa, 7.6, ± 5.3 GPa, 6.1 ± 3.8 GPa, 7.1 ± 4.2 GPa and 7.5 ± 2.5 GPa. For the toughness modulus, we measured respectively 71 ± 40, 65 ± 41, 60 ± 35, 70 ± 45 and 51 ± 39.

On the other hand, in Figure 2a and Appendix A, it is possible to notice a slight decrease of the diameter for longer samples. In particular, the diameter value for the 0.55 cm length was greater than the one for 0.75 cm (with a medium ES), and greater than the ones at 1.0 cm, 1.25 cm and 1.5 cm (with a large ES). Moreover, the diameter values for the samples 0.75 cm long were higher than those of the 1.0 cm long samples (with a medium ES). On the other hand, although the differences between the diameters of the samples at 0.75 cm and those at 1.25 cm and 1.5 cm were significative, their size effects were small. Respectively, we measured 5.5 ± 2.9 μm, 4.1 ± 1.6 μm, 3.2 ± 1.2 μm, 3.3 ± 1.9 μm and 3.4 ± 1.4 μm. This is in agreement with a recent work on the same species [35], which has highlighted that the dragline near the attachment disc (where the thread is attached) is composed of more fibers than the dragline far (in cm) from the attachment point, which thus has a smaller diameter. Appendix A shows optical and SEM images that highlight this feature, and could explain the observed difference. In fact, in spiders that fell greater distances (longer samples), it was unlikely that the dragline collected was near the attachment disc, i.e., had more fibers. 

The strength of the silk at different lengths is reported in Figure 2b (for the *p*-values and the Cohen’s coefficient, see Appendix A). Respectively, the values measured were 931 ± 345 MPa, 805 ± 371 MPa, 754 ± 315 MPa, 789 ± 317 MPa and 515 ± 260 MPa. We observed, thus, a slightly significant decrease in strength for the longer fibers. In particular, the fibers 0.55 cm long were significantly stronger with respect to those 1.0 cm long (with a medium ES) and those 1.5 cm long (with a very large ES). Moreover, the fibers 1.5 cm long were significantly weaker than those 0.75 cm, 1.0 cm and 1.25 cm long (with a large ES). 

In Figure 2c we plotted the linear regression method [Equation (3), see materials and methods] used to compute the Weibull parameters. All the data sets were acceptable to be dealt by means of Weibull statistics under the Kolmogorov–Smirnov test, and the obtained parameters were comparable with those obtained through the maximum likelihood method [36], confirming the applicability of Weibull statistics for our samples. In Table 3, these Weibull parameters are reported. Respectively, we obtained the following scale (*b*) and shape (*c*) parameters (in the brackets we inserted the parameters obtained through the maximum likelihood method): 1054 MPa and 2.7 (1044 MPa and 3.10), 909 MPa and 2.4 (910 MPa and 2.4), 860 MPa and 2.4 (849 MPa and 2.7), 894 MPa an 2.6 (889 MPa and 2.8) and 579 MPa and 2.3 (583 MPa and 2.1). The Weibull parameters that were obtained with the support of the linear regression method were used to plot the density probability function of the fracture strengths of the five different lengths (Figure 2d). These were similar, and no major differences appeared, indicating the similarity in statistical fracture behavior. Figure 2e represents the shape parameters obtained with the support of the linear regression method. No major differences appeared, indicating that Weibull statistics could be applied to this spider silk (i.e., the shape parameter can be considered constant). Figure 2f shows the decrease of the scale parameter [as predicted from Equation (6), see materials and methods, and the Weibull statistics] as a function of the length. Although the decrease is not big, we estimated the Weibull fractal dimension, using Equation (6), directly from the slope of the plot (i.e., −0.4584). We calculated *d* = 1.1, which is consistent with our prediction (*d* close to 1, due to the unidimensional nature of the silk fiber).

## 3. Discussion

The silks of nine species of spider were analyzed by means of mechanical tests. We found that the two distinct typologies of silk behaved differently from a statistical and mechanical point of view. One belongs to the infraorder of Mygalomorphae (Dipluridae and Theraphosidae families) and presented low shape parameters (between 1 and 2), which means that the fracture probability is more constant [8,30,31]. On the other hand, the Araneomorphae presented silks that were better in terms of mechanical performance, and had Weibull shape parameters higher than 2 (commonly found in other polymeric fibers, silk included [30,37]). 

The silk of the species *Cupiennius salei* was tested firstly at different strain rates. It being a polymeric viscoelastic material, the strain rate affects the capability of the silk to relax, and thus its mechanical properties [19,20,38,39]. In particular, we noticed a difference in terms of the mechanical properties, although the range of variation in the strain rates was relatively small, compared to that analyzed in a recent work [25]. Slightly higher values of Young’s modulus, strength and toughness modulus were found at 0.15 mm/s, suggesting that such a value is somewhat optimal, and is a figure compatible with natural conditions [19,40]. The improvement of such properties with increasing strain rates is in agreement with what has been found in the literature, here derived from a broader range of strain rates [25,41]. 

Together with the strain rates, we investigated how the length of the sample affects the mechanical properties of the silk (analysis performed on *Cupiennius salei* silk at five different lengths). We noticed a slightly significant decrease in the diameter for longer samples. This could be explained by the different numbers of fibers that compose the collected threads. It has been shown for the same species that draglines have a higher number of fibers when they are near the attachment discs [35], i.e., they have a greater diameter. This observation is relevant given the procedure of sample collection we employed; each sample being a consequence of a single fall, for spiders that fell greater distances it was unlikely that the dragline collected was near the attachment disc, i.e., had more fibers. At the same time, in the same work [35], it was also shown that these two distal portions of the same dragline have the same mechanical properties at a fixed length. Moreover, we found a slightly significant decrease in strength for longer threads, but not a significant difference in the other mechanical properties. This is due to the fact that the constitutive laws are highly nonlinear (Appendix A), e.g., the toughness modulus is not proportional to strength. 

As we expected, the consistency of the Weibull shape parameters for spider dragline silk at different lengths suggests that Weibull statistics could be applied to the strength of silk, and thus used to obtain its scaling laws. With the support of Equation (6), we plotted the Weibull scale parameters at different lengths, and computed the Weibull fractal dimension [31]. We obtained *d* = 1.1, which confirms the fact that defects in spider silk are proportional to the fiber length. This is in agreement with what was found through simulations and numerical works, where the failures of the silk occurred in the aligned crystalline regions along the pulling direction (i.e., traction tests) [42,43,44].

## 4. Material and Methods

### 4.1. Spiders and Silk

The spiders used in this work were kept in the lab under controlled environmental parameters. They were fed weekly with cockroaches or crickets. The silk was directly extracted from the spinnerets of the animals without the use of anesthetic. Each silk sample was obtained with a single silk extraction. In the case of dragline silk (major ampullate gland), it was sufficient to let the spider fall from a surface to induce the spinning of the thread, so for *n* samples the spider was let fall *n* times. For the spiders that belong to the families of Dipluridae and Theraphosidae, it was necessary to stimulate through contact the spinnerets with a tooth pick or tweezers. This operation was performed gently to avoid forcing the spinning, which affects the mechanical properties of the silk [20,45,46]. The extraction speed was around 1 cm/s. The investigated species were: *Araneus diadematus*, *Ancylometes* sp., *Ceratogyrus marshalli*, *Cupiennius*
*salei*, *Grammostola rosea*, *Linothele fallax*, *Nuctenea umbratica*, *Phoneutria fera* and *Zygiella x-notata*. These species were selected because of the availability of adult females in our farm. *Cupiennius salei* was chosen because it is a massive wandering spider that produces easily collectable dragline. For the measurement of the silk at different strain rates, we used a different spider (*Cupiennius salei*) with respect to the ones used to measure the strength at different lengths. 

### 4.2. Tensile Tests

The extracted silk was glued, with double sided tape, on a paper support (square window with side of 1 cm) by following the procedure of Blackledge et al. [38]. The measure of the diameter was done with the support of an optical microscope and ImageJ [47]. The shape of the threads was assumed to be circular. The samples were mounted on a nanotensile machine “Agilent T150 UTM”. For the tensile tests performed at different strain rates, these were 0.08 mm/s, 0.10 mm/s, 0.11 mm/s, 0.15 mm/s and 0.17 mm/s. On the other hand, for the tensile tests performed at different lengths, the strain rate was 0.11 mm/s. The lengths were 0.55 cm, 0.75 cm, 1.0 cm, 1.25 cm and 1.5 cm. The engineering strain was obtained by dividing the displacement for the gauge length. The Young’s modulus was computed by the slope of the stress strain curve in the initial linear elastic part. The toughness modulus was obtained by measuring the area under the stress strain curve. Finally, the ultimate strength was obtained by taking the final engineering stress prior to fracture. All silk threads were tested two weeks after their production to stabilize the concentration of residual stresses and thus to minimize their effects on the mechanical properties [48,49,50]. All samples were kept in controlled conditions (20–21 °C and 39–42% RH) during this period to create uniform testing conditions. At least 15 samples per species and data set were measured. For the silk of *Cupiennius salei* at 5 different speeds and 5 different lengths we tested at least 27 samples per data set. 

### 4.3. Weibull Statistics

Nine sets of ultimate strength values were measured from the nine species of spiders analyzed. For each set we estimated Weibull parameters using the linear regression method. By applying to Equation (1) the double logarithm we obtain
(3)$$\ln\left(\ln\left(\frac{1}{1-F\left(x\right)}\right)\right)=c\ln\left(x\right)-c\ln\left(b\right)$$
where *x* is the experimental value. After the organization of the values from the lowest to the biggest, *F*(*x*) was computed with estimator median rank [51,52]
(4)$$\hat{F}\left(x_{i}\right)=\frac{i-0.3}{n+0.4}$$
where *i* is the position of the *i*-th value and *n* is the dimension of the sample. $\hat{F}\left(x_{i}\right)$ is the cumulative density function associated to each value of ultimate stress only by considering its relative position. The Equation (3) was plotted for each set of data, where the shape parameter (Weibull modulus *c*) was simply computed as the slope of the plot and the scale parameter (*b*) with this expression
(5)$$b=e^{-\left(\frac{\gamma }{c}\right)}$$
where *γ* is the intercept on the vertical axis (namely −cln(b)).

We performed the Kolmogorov Smirnov test to verify that for each set of data we could applied Weibull’s statistics with the obtained parameters. An acceptance value of 95% was taken (Matlab^®^, The MathWorks, Natick, MA, USA). Moreover, for each data set we used the maximum likelihood principle to investigate if the Weibull’s parameters were consistent with respect to the number of tested samples (following Peterlick et al. [36] and by means of Mathemathica^®^ software) (Wolfram Research, Inc., Champain, IL, USA).

To evaluate the scaling laws and the Weibull fractal dimension *d* of the silk of the species *Cupiennius salei*, we used five sizes: 0.55 cm, 0.75 cm, 1.0 cm, 1.25 cm and 1.5 cm. The shape parameter *m* and the constant *b* were computed for each data set (linear regression method). The analysis of the scaling law and Weibull fractal dimension was performed using Equation (2). We applied to it the logarithm as:(6)$$\ln\left(b\right)=-\frac{d}{c}\ln\left(l\right)+\ln\left(x_{0}l_{0}^{\frac{d}{c}}\right)$$
from where we can obtain directly the value of *d* from the slope of this plot and by knowing the value of the shape parameter *c*, which is theoretically constant by changing the length of the sample. In this context, we used the average of *c* obtained among all the *Cupiennius salei* length data set. 

### 4.4. ANOVA Analysis

One-way Analysis of Variance (pairwise comparison) was performed to compare the mechanical properties of the fibers obtained under different parameters. That is, one-way ANOVA enables us to find out whether the two groups of an independent variable (sample length, strain rate) have different effects on the response variables, i.e., strain at break, strength, Young’s modulus, and toughness modulus.

The parameters used to verify the null hypothesis, i.e., all the data sets come from the same distribution and have the same mean value, were
(7)$$SSQ_{a}=\overset{G}{\underset{g=1}{\sum }}n_{g}{\left(m_{g}-m_{u}\right)}^{2}$$
(8)$$SSQ_{e}=\overset{G}{\underset{g=1}{\sum }}\overset{n_{g}}{\underset{j=1}{\sum }}{\left(x_{gj}-m_{g}\right)}^{2}$$
where *G* is the number of different samples under consideration, *n_g_* is the number of tests of the same sample, *m_u_* is the mean value of all the data, *m_g_* is the mean value within the group (i.e., sample), and *x* is the single quantity value. These sums of squares were used to compute the *T* value, as:(9)$$T=\frac{\frac{SSQ_{a}}{G-1}}{\frac{SSQ_{e}}{n-G}}$$
which has been compared with the ideal value of the Fisher function *F_F_*, with a significance level of 5%. If *T* > *F_F_*, we reject the null hypothesis and thus we can consider the difference among the data set as significant. In this case, the null hypothesis is that the differences among the mean values of five groups of the independent variable (sample length or strain rate) are consequences of the internal variance in the groups, and thus they are not due to an intrinsic difference. The test function *T* and the two-tailed *p*-value were computed with the support of Matlab^®^.

### 4.5. Effect Size (ES)

The *p*-value is affected by the sample’s dimension. Larger samples are likely to give statistically significant differences, even though such differences are small [53]. For this reason, it is important to consider the ES, which is a measure of the magnitude of such difference [54]. We based our analysis on the parameters introduced by Cohen [54]. By assuming that the two compared populations have the same variance, a pooled standard deviation can be defined as
(10)$$s=\sqrt{\frac{\left(n_{1}-1\right)s_{1}^{2}+\left(n_{2}-1\right)s_{2}^{2}}{n_{1}+n_{2}-2}}$$
where *n*_1_ and *n*_2_ are the dimensions of the two groups compared, and *s*_1_ and *s*_2_ their standard deviations. With *s*, Cohen defined the following parameter
(11)$$d_{c}=\frac{m_{1}-m_{2}}{s}$$
where *m*_1_ and *m*_2_ are respectively the means of the two groups. Based on Cohen [54] and Sawiloski [44], *d_c_* helps to define qualitatively the magnitude of the difference of the means as very small (*d_c_* ≥ 0.01, i.e., circa 100% distributions’ overlap), small (*d_c_* ≥ 0.20, i.e., circa 85% distributions’ overlap), medium (*d_c_* ≥ 0.50, i.e., circa 67% distributions’ overlap), large (*d_c_* ≥ 0.8, i.e., circa 53% distributions’ overlap), very large (*d_c_* ≥ 1.20, i.e., circa 40% distributions’ overlap), and huge (*d_c_* ≥ 2.0, i.e., circa 19% distributions’ overlap). 

## 5. Conclusions

The understanding of the mechanical properties of spider silks is crucial per se, and also for its use in bioinspired and biocompatible applications. In this work, we studied the silks of nine species of spider (some of them unknown in the literature) by measuring their mechanical properties. Moreover, for the species *Cupiennius salei*, we tested the silk at different strain rates and lengths. For these, we computed the Weibull parameters and analyzed their scaling law. 

These results could improve the applicability of spider silks to novel technologies. Moreover, they offer a protocol for performing a statistical analysis of native or artificial spider silks and the related systems.

## Figures and Tables

**Figure 1 molecules-25-02938-f001:**
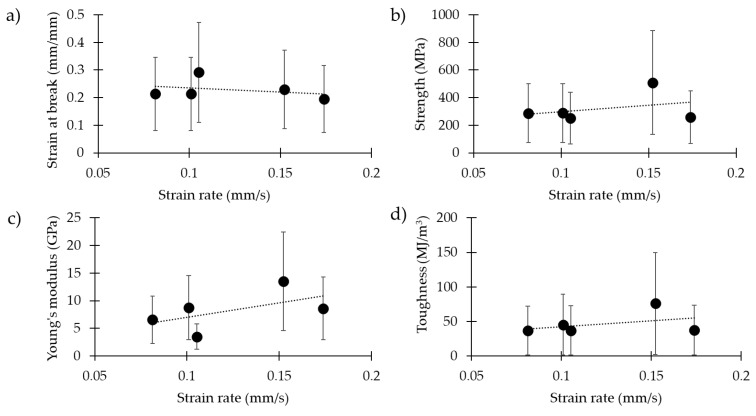
Mechanical properties of the spider silk of *Cupiennius salei* at different strain rates: (**a**) strain at break, (**b**) strength, (**c**) Young’s modulus, and (**d**) toughness modulus.

**Figure 2 molecules-25-02938-f002:**
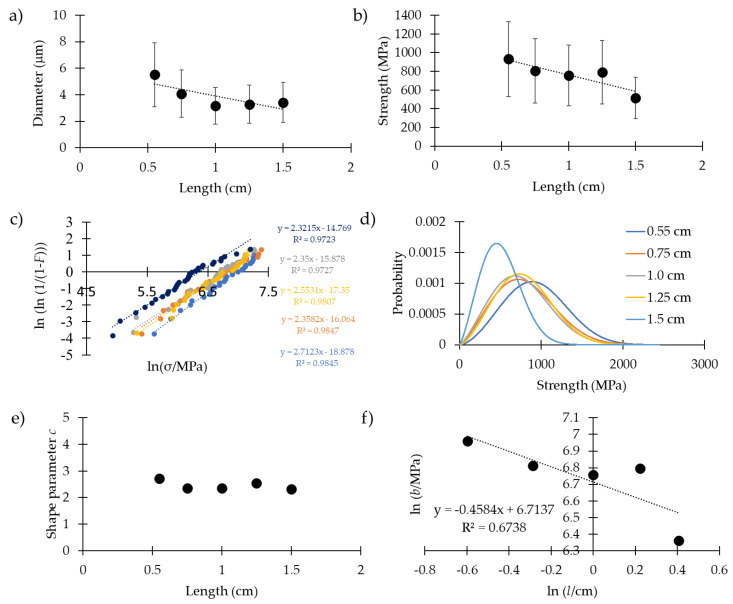
(**a**) The diameter of the dragline vs the length of the sample. (**b**) Strength of the dragline vs the length of the sample. (**c**) Linear regression plot of the strength data set used to compute the Weibull parameters. (**d**) Weibull’s probability density distribution of the strength at different lengths. (**e**) Shape parameter vs length of the sample. (**f**) Plot of Equation (6) with linear regression to compute the Weibull fractal dimension.

**Table 1 molecules-25-02938-t001:** Mechanical properties and Weibull parameters of different spider silks.

Species	Nr. Samples	Strain at Break (mm/mm)	Strength (MPa)	Young’s Modulus (GPa)	Toughness Modulus (MJ/m^3^)	Scale Parameter *b* (MPa)	Shape Parameter (*c*)
*Araneus diadematus*	16	0.25 ± 0.09	655 ± 286	8.5 ± 4.9	267 ± 164	949	2.56
*Ancylometes bogotensis*	15	0.24 ± 0.08	897 ± 441	21.8 ± 9.9	191 ± 175	1112	2.01
*Ceratogyrus marshalli*	15	0.17 ± 0.15	163 ± 158	2.6 ± 2.1	14 ± 11	236	1.16
*Cupiennius salei*	36	0.29 ± 0.15	253 ± 217	3.5 ± 2.8	37 ± 35	859	2.35
*Grammostola rosea*	15	0.17 ± 0.15	13 ± 9	3.0 ± 2.2	26 ± 16	42	1.58
*Linothele fallax*	15	0.27 ± 0.16	110 ± 86	5.0 ± 4.3	21 ± 20	127	1.22
*Nuctenea umbratica*	15	0.21 ± 0.06	1199 ± 725	10.2 ± 4.2	138 ± 81	1693	2.35
*Phoneutria fera*	15	0.32 ± 0.19	936 ± 544	27.2 ± 13	202 ± 141	1191	3.14
*Zygiella x-notata*	15	0.19 ± 0.07	283 ± 137	5.0 ± 2.6	36 ± 25	597	2.19

**Table 2 molecules-25-02938-t002:** Mechanical properties of the dragline silk at different strain rates.

Strain Rate (mm/s)	Nr. Samples	Diameter (μm)	Strain at Break (mm/mm)	Strength (MPa)	Young’s Modulus (GPa)	Toughness Modulus (MJ/m^3^)
0.08	33	3.5 ± 1.5	0.21 ± 0.15	288 ± 241	6.6 ± 3.9	36 ± 41
0.10	37	4.0 ± 1.2	0.21 ± 0.12	289 ± 218	8.8 ± 6.5	45 ± 46
0.11	36	3.3 ± 0.9	0.29 ± 0.15	253 ± 217	3.5 ± 2.8	37 ± 35
0.15	31	3.0 ± 1.6	0.23 ± 0.17	510 ± 311	13.5 ± 6.7	76 ± 63
0.17	35	3.9 ± 1.5	0.20 ± 0.11	259 ± 168	8.6 ± 5.4	37 ± 36

**Table 3 molecules-25-02938-t003:** Mechanical properties and Weibull parameters of the dragline silk at different lengths. In italics and between brackets are the Weibull parameters obtained through the maximum likelihood method, following Peterlik [36].

Length (cm)	Nr. Samples	Diameter (μm)	Strain at Break (mm/mm)	Young’s Modulus (GPa)	Toughness Modulus (MJ/m^3^)	Strength (MPa)	Shape Parameter *c*	Scale Parameter *b* (MPa)
0.55	29	5.5 ± 2.9	0.23 ± 0.11	8.0 ± 4.1	39 ± 40	932 ± 345	2.7 (*3.10*)	1054 (1044)
0.75	29	4.1 ± 1.6	0.20 ± 0.15	7.6 ± 5.3	45 ± 41	805 ± 371	2.4 (*2.4*)	909 (910)
1.0	27	3.2 ± 1.2	0.25 ± 0.14	6.1 ± 3.8	60 ± 43	754 ± 315	2.4 (*2.7*)	860 (849)
1.25	28	3.3 ± 1.9	0.22 ± 0.10	7.1 ± 4.2	51 ± 45	790 ± 317	2.6 (*2.8*)	894 (889)
1.5	33	3.4 ± 1.4	0.21 ± 0.17	7.5 ± 2.5	64 ± 39	515 ± 260	2.3 (*2.1*)	579 (583)

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
