# Peer review of "Mechanical Properties and Weibull Scaling Laws of Unknown Spider Silks"

_molecules, 2020, doi:10.3390/molecules25122938_

Round 1
Reviewer 1 Report
In this short paper the authors report on investigations of the mechanical properties of several spider silks, including some that had not been previously reported. In particular, the authors also studied the materials in terms of Weibull statistics, and the authors investigated the mechanical properties as a functions of different strain rates.
It is a bit unfortunate that the language quality of the paper is not very high. Mostly, these errors appear to be sloppy mistakes and typos, such as nouns in plural when they should be singular, and vice versa. There are also some extra words that look like leftovers from previous editing iterations, such as the word "strength" in this sentence, which does not seem to fit in (L. 223-225): "In this work we have studied the silk of nine species (some of them unknown in literature) by measuring the stress-strain curves and obtaining their mechanical properties strength of their silk." Also the words "of their silk" at the end of the sentence seem to be redundant.
One major question about this manuscript is about the statistics. Although the authors detail in the methods section how the statistics analysis was done, several major questions remain. Not surprisingly for mechanical tests of microscopically small fiber samples, the spread of the data in all parameters (stiffness, strength, strain at break, toughness) is significant. Despite to this large spread, the authors claim that the comparatively relatively small changes in these 4 parameters as a function of strain rate are extremely highly statistically significant with extremely small p-values on the order of 0.00002. Looking at the large error bars, this appears to be quite unreasonable. It is not clear to the reviewer how these results were obtained. For instance, the authors claim that the strain at break significantly decreased as a function of increasing strain rate. While such a conclusion is not surprising in general, it is not clear how the authors derived it from their data. How does one define a null hypothesis for this scenario? While it is quite straightforward for two populations, in this case, there were five groups of results (five strain rates). Was the goal to prove that each population at a given strain rate had a strain at break higher than all other populations at lower strain rates (strictly monotonically growing)? This needs to be explained in detail, especially since the conclusions do not appear to be sound just by looking at the data (which is admittedly subjective). Especially it appears that most of the mentioned trends are caused by the results acquired at 0.15 mm/s, which are higher than the other ones, albeit at significantly larger standard deviations.
Did the authors calculate one-sided (one-tailed) or two-sided (two-tailed) p-Values? In this case, it is hard to see how a one-tailed approach would be sufficient, since it is not clear a priori whether any of the measured properties would increase or decrease as a function of strain rate.
Also, the authors state that (L.241) that the length of the sample only affects strength but not the other properties. How can the strength be affected without affecting the toughness modulus (or strain at break)?
Author Response
1) “In this short paper the authors report on investigations of the mechanical properties of several spider silks, including some that had not been previously reported. In particular, the authors also studied the materials in terms of Weibull statistics, and the authors investigated the mechanical properties as a functions of different strain rates. It is a bit unfortunate that the language quality of the paper is not very high. Mostly, these errors appear to be sloppy mistakes and typos, such as nouns in plural when they should be singular, and vice versa. There are also some extra words that look like leftovers from previous editing iterations, such as the word "strength" in this sentence, which does not seem to fit in (L. 223-225): "In this work we have studied the silk of nine species (some of them unknown in literature) by measuring the stress-strain curves and obtaining their mechanical properties strength of their silk." Also the words "of their silk" at the end of the sentence seem to be redundant.”
Reply:
We thank the reviewer for this comment. We have improved the English grammar in all the text of the revised version e.g. removing “strength of their silk” from the mentioned sentence (see marked copy).
2) “One major question about this manuscript is about the statistics. Although the authors detail in the methods section how the statistics analysis was done, several major questions remain. Not surprisingly for mechanical tests of microscopically small fiber samples, the spread of the data in all parameters (stiffness, strength, strain at break, toughness) is significant. Despite to this large spread, the authors claim that the comparatively relatively small changes in these 4 parameters as a function of strain rate are extremely highly statistically significant with extremely small p-values on the order of 0.00002. Looking at the large error bars, this appears to be quite unreasonable.”
Reply:
This is a consequence of how ANOVA defines the p-value. That is, one-way ANOVA has enabled us to find out whether different groups (which can be more than two, i.e. in this case 5 groups) of an independent variable (sample length, strain rate speed) had different effects on the response variable (strain at break, strength, Young’s modulus, toughness modulus). In this context, to compute the p-value the T-test function must be considered. In doing so, one has to divide the sum of squares (relative to the variation within a group) for the total number of cases (Equation 9). We analysed a big number of cases (146 for the lengths and 172 for the strain rates) that had a huge effect on the p-values.
We have accordingly added/modified the following sentences:
Line 107:
“(including the out layers)”
Lines 139-141:
“That is, one-way ANOVA enables us to find out whether the five groups of an independent variable (sample length, strain rate) have different effects on the response variables, i.e. strain at break, strength, Young’s modulus, and toughness modulus.”
Lines 152-154:
“In this case the null hypothesis is that the differences among the mean values of five groups of the independent variable (sample length or strain rate) are consequences of the internal variance in the groups, thus they are not due to an intrinsic difference.”
Lines 277-278:
“These findings were based on the p-value, which was calculated among all the 5 groups and thus affected by the large number of samples (172).”
Lines 293-295:
“As in the case of different strain rates, this finding is based on the p-value, which was calculated among all the 5 groups and thus affected by the large number of samples (146).”
3) “It is not clear to the reviewer how these results were obtained. For instance, the authors claim that the strain at break significantly decreased as a function of increasing strain rate. While such a conclusion is not surprising in general, it is not clear how the authors derived it from their data.“
Reply:
After we found that the difference among the groups was significant (see the answer to the previous comment), we added to the graphs the trend lines that indicates the direction of change, i.e. if there is a decrease or an increase.
We have accordingly added the following sentences:
Lines 156-158:
“Finally, in order to evaluate the direction of change (i.e. increase or decrease), a trend line has been plotted by considering all the data.”
Line 181:
“This was stated by looking at the trend line (Figure 1a).”
Line 184:
“(see trend line in Figure 1b)”
Line 187-188:
“(see trend line in Figure 1c)”
Line 190:
“(see trend line in Figure 1d)”
4) “How does one define a null hypothesis for this scenario?”
Reply:
The null hypothesis is defined as: “The differences among the mean values of five groups of an independent variable (sample length, strain rate) are consequences of the internal variance in the groups, thus they are not due to an intrinsic difference.”
We have accordingly added the following lines:
Lines 152-154:
“In this case the null hypothesis is that the differences among the mean values of five groups of the independent variable (sample length or strain rate) are consequences of the internal variance in the groups, thus they are not due to an intrinsic difference.”
5) “While it is quite straightforward for two populations, in this case, there were five groups of results (five strain rates). Was the goal to prove that each population at a given strain rate had a strain at break higher than all other populations at lower strain rates (strictly monotonically growing)?”
Reply:
We have compared the 5 populations according to the null hypothesis. The goal was to verify that they had different mean values (i.e. the groups were different). In fact, ANOVA does not give us the direction of change and so we plotted the trend line that is informative about this, i.e. if there was a decrease of an increase.
6) “This needs to be explained in detail, especially since the conclusions do not appear to be sound just by looking at the data (which is admittedly subjective). Especially it appears that most of the mentioned trends are caused by the results acquired at 0.15 mm/s, which are higher than the other ones, albeit at significantly larger standard deviations.”
Reply:
We have detailed the previous points as previously reported whereas concerning the 0.15mm/s strain rates, it is a strong contributor to the slope of the trend line, as well as the other 4 groups. What we measured and computed is an average behaviour. The slope of the trend line is computed by plotting all the values. The highly different values at 0.15 mm/s have thus particularly affected such values. We have accordingly added the following lines.
Lines 193-194:
“Moreover, we highlight the fact that the highly different values at 0.15 mm/s have particularly affected the slope of the trend line.”
Lines 274-276:
“(often maximum values of the mechanical properties and particularly Young’s modulus were found at 0.15 mm/s suggesting that such a value is a kind of optimal and in number compatible with natural conditions6,7)”
7) “Did the authors calculate one-sided (one-tailed) or two-sided (two-tailed) p-Values? In this case, it is hard to see how a one-tailed approach would be sufficient, since it is not clear a priori whether any of the measured properties would increase or decrease as a function of strain rate.”
Reply:
We performed one-way ANOVA analysis with two tailed p-values (standard protocol by using Matlab®). Concerning the mentioned a priori knowledge we expected the trend measured (see ref Yazawa et al.1) even though we did not know if in our small interval of strain rates we would have been able to detect any significant difference. For the sake of clarity, we added/modified the following lines:
Lines 155-156:
“The test function T and the two-tailed p-value were computed with the support of Matlab®.”
8) “Also, the authors state that (L.241) that the length of the sample only affects strength but not the other properties. How can the strength be affected without affecting the toughness modulus (or strain at break)?”
Reply:
We thank the reviewer for this question. The constitutive laws are highly not linear, we have now added representative stress-strain curves in the Supplementary Materials, thus toughness is not proportional to strength. This is the reason why ANOVA test did not provide a significant difference for the other properties. We have accordingly added/modified the following lines:
Lines 160-162:
“The mechanical properties and Weibull parameters of nine species of spiders were obtained by means of their stress-strain curves (Figure S1, Table 1), which showed the typical nonlinear constitutive law of the silk11.”
Lines 290-293:
“Moreover, we found a slightly significant decrease in strength for longer threads but not a significant difference in the other mechanical properties. This is due to the fact that the constitutive laws are highly nonlinear (Figure S1), e.g. toughness modulus is not proportional to strength. This is the reason why ANOVA test did not provide a significant difference for the other properties.”
Added references:
- Yazawa, K., Malay, A. D., Masunaga, H., Norma-Rashid, Y. & Numata, K. Simultaneous effect of strain rate and humidity on the structure and mechanical behavior of spider silk. Commun. Mater. 1–10 (2020). doi:10.1038/s43246-020-0011-8
- Pérez-Rigueiro, J., Elices, M., Llorca, J. & Viney, C. Tensile properties of Argiope trifasciata drag line silk obtained from the spider’s web. J. Appl. Polym. Sci. 82, 2245–2251 (2001).
- Rinne, H. The Weibull Distribution. A handbook. (2009).
- Prez-Rigueiro, J., Elices, M. & Llorca, C. V. Tensile properties of Argiope trifasciata drag line silk obtained from the spider’s web. J. Appl. Polym. Sci. 82, 2245–2251 (2001).
- Peterlik, H. The validity of Weibull estimators. J. Mater. Sci. 30, 1972–1976 (1995).
- Denny, M. W. the Physical Properties of Spider’S Silk and Their Role in the Design of Orb-Webs. J. Exp. Biol. 65, 483–506 (1976).
- Ortlepp, C. & Gosline, J. M. The scaling of safety factor in spider draglines. J. Exp. Biol. 211, 2832–2840 (2008).
- Greco, G., Wolff, J. & Pugno, N. M. Strong and tough silk for resilient attachment discs: the mechanical properties of piriform silk, in the spider Cupiennius salei (Keyserling, 1877). Front. Mater. In press., (2020).
- Swanson, B. O., Blackledge, T. A., Beltrán, J. & Hayashi, C. Y. Variation in the material properties of spider dragline silk across species. Appl. Phys. A Mater. Sci. Process. 82, 213–218 (2006).
- Swanson, B. O., Anderson, S. P., DiGiovine, C., Ross, R. N. & Dorsey, J. P. The evolution of complex biomaterial performance: The case of spider silk. Integr. Comp. Biol. 49, 21–31 (2009).
- Agnarsson, I., Kuntner, M. & Blackledge, T. A. Bioprospecting finds the toughest biological material: Extraordinary silk from a giant riverine orb spider. PLoS One 5, 1–8 (2010).
Reviewer 2 Report
The authors investigated the effects of strain rate and fiber length on the mechanical property of spider silks, including the silks whose mechanical property has never been reported. They detected significant effects in both strain rate and fiber length on the mechanical property. In addition, they showed that the effect of fiber length on the strength follows Weibull scaling laws. The obtained conclusion is interesting and valuable not only in the silk science community but also in the communities of other kinds of ductile fibers in both natural and synthetic fibers. So, if the following requirements are satisfied, it would be acceptable for Molecules.
- Weibull analysis is useful essentially for brittle materials, in which the constituent structural elements are not deformed significantly. On the other hand, the silks, including spider silks, are representative tough and ductile materials, in which the constituent structural elements can be highly deformed during the stretching. It is also known that the deformation of such structural elements occurs every second during the stretching and the deformation leads to the changes of hierarchical structure (or fiber structure) and directly affects on the stress-strain curve. So, the authors should explain, probably in both Introduction and Discussion parts, the validity for applying Weibull analysis into the break of such ductile spider silks.
- Related to Comment 1, I request the authors to explain more concretely about the characteristic length “l0” in equation (1). Which kind of length do you suppose for your silk samples?
- Figure 1b, c, d: The authors said, significant increases in strength, modulus, and toughness were observed with increasing the testing speed. However, it looks also that the maximum strain rate is there (at 0.15mm/s). How do you think about it?
- Figure 2a and Lines186-191: The authors said, a lightly decrease of the diameter with respect to the length probably due to the different stretching history depending on the length. Is this consideration reasonable? The sentences make me imagine that you collected each sample, having different length, independently. However, I guess that you collected certain length of long thread first and then later you cut it into each sample length. If so, the stretching history of all the samples should be same. The sentences also make me imagine that the different stretching history gives different hierarchical structure between samples with different length. So, it is required to explain more concrete about these point.
- Can you add into Figure 2 one more sample-point of much longer sample length ca. 10cm or 20cm? This adding may make more clear your conclusion.
Thats all.
Author Response
1) “The authors investigated the effects of strain rate and fiber length on the mechanical property of spider silks, including the silks whose mechanical property has never been reported. They detected significant effects in both strain rate and fiber length on the mechanical property. In addition, they showed that the effect of fiber length on the strength follows Weibull scaling laws. The obtained conclusion is interesting and valuable not only in the silk science community but also in the communities of other kinds of ductile fibers in both natural and synthetic fibers. So, if the following requirements are satisfied, it would be acceptable for Molecules.”
Reply:
We thank the reviewer for this positive comment.
2) “Weibull analysis is useful essentially for brittle materials, in which the constituent structural elements are not deformed significantly. On the other hand, the silks, including spider silks, are representative tough and ductile materials, in which the constituent structural elements can be highly deformed during the stretching. It is also known that the deformation of such structural elements occurs every second during the stretching and the deformation leads to the changes of hierarchical structure (or fiber structure) and directly affects on the stress-strain curve. So, the authors should explain, probably in both Introduction and Discussion parts, the validity for applying Weibull analysis into the break of such ductile spider silks.”
Reply:
Weibull statistics was derived in the context of brittle materials, but now is widely used for different materials and different properties, e.g. see refs Perez-Rigueiro et al.2, and Rinne3. The validity of such statistics can only be checked afterwards as we did, by using Kolmogorov Smirnov test (to quantify the validity of the fits) coupled with the maximum likelihood principle (to see the variation of the computed Weibull parameters).
In order to make this point clearer, we added/modified the following lines:
Lines 49-51:
“In this context, Weibull statistics has been widely used in past and recent years to describe the statistical distribution of the ultimate strength of many natural or artificial materials (not necessarily brittles) including the silk of the spider Argiope trifasciata3,4”
Lines 223-225:
“All the datasets were acceptable for Weibull statistics under Kolmogorov Smirnov test and the obtained parameters were comparable with those obtained through maximum likelihood method5, confirming the applicability of Weibull statistics for our samples.”
Lines 296-298:
“As we expected, the constancy of the Weibull shape parameters for spider dragline silk at different length suggests that Weibull statistics could be applied to the strength of silk and thus used to obtained its scaling laws.”
3) “Related to Comment 1, I request the authors to explain more concretely about the characteristic length “l0” in equation (1). Which kind of length do you suppose for your silk samples?”
Reply:
This length is arbitrary, only has a physical meaning by governing the mean value of the Weibull distribution, however it is commonly introduced to make the equation dimensionless.
4) “Figure 1b, c, d: The authors said, significant increases in strength, modulus, and toughness were observed with increasing the testing speed. However, it looks also that the maximum strain rate is there (at 0.15mm/s). How do you think about it?”
Reply:
The fact that at 0.15 mm/s we observed the maximum in our 5 setups is interesting and we agree with the Referee that has to be emphasized. We have thus remarked it adding the following sentence:
Lines 274-276:
“(often maximum values of the mechanical properties and particularly Young’s modulus were found at 0.15 mm/s suggesting that such a value is a kind of optimal and in number compatible with natural conditions6,7)”
5) “Figure 2a and Lines186-191: The authors said, a lightly decrease of the diameter with respect to the length probably due to the different stretching history depending on the length. Is this consideration reasonable? The sentences make me imagine that you collected each sample, having different length, independently. However, I guess that you collected certain length of long thread first and then later you cut it into each sample length. If so, the stretching history of all the samples should be same. The sentences also make me imagine that the different stretching history gives different hierarchical structure between samples with different length. So, it is required to explain more concrete about these point.”
Reply:
Each sample was collected separately by letting the spider fall down. So, if we measured n samples, the spider fell n times. The stretching history of the samples was assumed the same since their rest for more than two weeks in the testing room (see materials and methods). We agree that we phrased wrongly the sentence and we think that this is an interesting point to be discussed. In an our recent work on the same species8, we noticed that the dragline near the attachment disc (where the thread is attached) is composed of more fibres than the dragline far from the attachment point. We believe that the reason why we could have measured a small decrease in diameter is that if the spider fall for longer distances it is unlikely that the dragline collected is near the attachment disc, i.e. has more fibres. In the same work, we demonstrated that there is no significant difference in mechanical properties at a fixed length of these two types of dragline.
In order to clarify this point, we added a figure (optical and SEM images of the two types of dragline) in the supplementary information and we added/modified the following lines:
Lines 76-78:
“Each silk sample was obtained with a single silk extraction. In the case of dragline silk (major ampullate gland) it was sufficient to let the spider fall from a surface to induce the spinning of the thread, so for n samples the spider was let fall n times.”
Lines 212-217:
“This is in agreement with a recent work on the same species8, which has highlighted that the dragline near the attachment disc (where the thread is attached) is composed of more fibres than the dragline far (some cm) from the attachment point, which has thus a smaller diameter. Figure S2 shows optical and SEM images that highlight this feature and could explain the observed difference. In fact, in spiders that fell for longer distances (longer samples) it was unlikely that the dragline collected was near the attachment disc, i.e. has more fibres.”
Lines 282-290:
“We noticed a slightly significant decrease in the diameter in longer samples. This could be explained by noticing the different number of fibres that compose the collected threads. It has been shown for the same species that draglines have a higher number of fibres near the attachment discs8, i.e. bigger diameter. This is related to our observation because of the procedure of samples’ collection. In fact, being each sample a consequence of a single fall, in spiders that fell for longer distances it was unlikely that the dragline collected was near the attachment disc, i.e. had more fibres. At the same time, in the same work8 it has also been shown that these two distal portions of the same dragline have the same mechanical properties at fixed length.”
6) “Can you add into Figure 2 one more sample-point of much longer sample length ca. 10cm or 20cm? This adding may make more clear your conclusion.
Thats all.”
Reply:
Unfortunately, due to the current situation we cannot access the laboratory and we cannot perform more experiments. We are actually thinking about extending this analysis on large scale, but this is planned to be a future work.
Added references:
- Yazawa, K., Malay, A. D., Masunaga, H., Norma-Rashid, Y. & Numata, K. Simultaneous effect of strain rate and humidity on the structure and mechanical behavior of spider silk. Commun. Mater. 1–10 (2020). doi:10.1038/s43246-020-0011-8
- Pérez-Rigueiro, J., Elices, M., Llorca, J. & Viney, C. Tensile properties of Argiope trifasciata drag line silk obtained from the spider’s web. J. Appl. Polym. Sci. 82, 2245–2251 (2001).
- Rinne, H. The Weibull Distribution. A handbook. (2009).
- Prez-Rigueiro, J., Elices, M. & Llorca, C. V. Tensile properties of Argiope trifasciata drag line silk obtained from the spider’s web. J. Appl. Polym. Sci. 82, 2245–2251 (2001).
- Peterlik, H. The validity of Weibull estimators. J. Mater. Sci. 30, 1972–1976 (1995).
- Denny, M. W. the Physical Properties of Spider’S Silk and Their Role in the Design of Orb-Webs. J. Exp. Biol. 65, 483–506 (1976).
- Ortlepp, C. & Gosline, J. M. The scaling of safety factor in spider draglines. J. Exp. Biol. 211, 2832–2840 (2008).
- Greco, G., Wolff, J. & Pugno, N. M. Strong and tough silk for resilient attachment discs: the mechanical properties of piriform silk, in the spider Cupiennius salei (Keyserling, 1877). Front. Mater. In press., (2020).
- Swanson, B. O., Blackledge, T. A., Beltrán, J. & Hayashi, C. Y. Variation in the material properties of spider dragline silk across species. Appl. Phys. A Mater. Sci. Process. 82, 213–218 (2006).
- Swanson, B. O., Anderson, S. P., DiGiovine, C., Ross, R. N. & Dorsey, J. P. The evolution of complex biomaterial performance: The case of spider silk. Integr. Comp. Biol. 49, 21–31 (2009).
- Agnarsson, I., Kuntner, M. & Blackledge, T. A. Bioprospecting finds the toughest biological material: Extraordinary silk from a giant riverine orb spider. PLoS One 5, 1–8 (2010).
Reviewer 3 Report
Gabriele Greco and Nicola M. Pugno presented a systematic and statistical investigation on the mechanical properties of a considerable number of different spider silks, some of them never reported in literature. Silks, and specially spider silks, are very attractive materials with excellent mechanical properties. Besides that, also present characteristics as biocompatibility, biodegradability, capacity to interact with functional molecules. In the last years, several innovative works using silk were present in top areas such as biomedicine, optics, energy or sensors. The data presented in this work can add valuable information for future works.
I propose minor revisions, mainly related to text formatting, and propose some suggestions to the authors:
The title of this manuscript is misleading. Are the mechanical properties or the spiders the “unknown” part? Besides that, the authors presented data of nine species of spiders, but only the data of 4 are absent in literature.
Page 2, line 80: “Fera” should be changed to “fera”
Pag 2, line 88: “second” should be changed to “s”
Several text spaces are missing or wrong along all the manuscript. For example, pag 2, line 88: “1% /second” or “0.1mm/s”; or pag 4, line 142: “T>F”. Please revise this along all the manuscript.
Page 4, line 158. Remove the sentence “The mechanical properties of the silks of Ancylometes bogotensis, Ceratogyrus marshalli, Linothele fallax, and Phoneutria fera were not reported in literature.” from the caption of Table 1. There is no need to have this information in the caption. Beside that, its already in the line 148-149.
All the graphics should be revised. Minor and major ticks in the xx and yy axis’s are missing and the graphics should be redesigned to be visually more attractive and easy to analyze data.
Pag. 8, line 223: Is there any substantial reason to have “Bioinspired” and “Bio-compatible” with the first letter capitalized? “Bio-compatible” should be changed to “biocompatible”.
Pag. 8, line 223 “In this work we have studied the silk of nine species (some of them unknown in literature) by measuring the stress-strain curves and obtaining their mechanical properties strength of their silk.”. It must be rewritten or “…of their silk” eliminated.
Page 8, line 230: “The former presented low Weibull modulus (between 1 and 2) maintaining constant in time fracture probability and thus mechanical behavior7,29,30 (comparable to what can be found in literature on this types of silks7).” This sentence is confused, it must be rewritten. “...this types…” should be “…these types…”.
Are any of the presented data surprising for the authors?
How and why were these spider species chosen for this study?
The presented data (not only the mechanical behavior) should be compared with the ones of the species that are already available in literature.
Author Response
1) “Gabriele Greco and Nicola M. Pugno presented a systematic and statistical investigation on the mechanical properties of a considerable number of different spider silks, some of them never reported in literature. Silks, and specially spider silks, are very attractive materials with excellent mechanical properties. Besides that, also present characteristics as biocompatibility, biodegradability, capacity to interact with functional molecules. In the last years, several innovative works using silk were present in top areas such as biomedicine, optics, energy or sensors. The data presented in this work can add valuable information for future works.”
Reply:
We thank the reviewer for this positive comment.
2) “I propose minor revisions, mainly related to text formatting, and propose some suggestions to the authors:The title of this manuscript is misleading. Are the mechanical properties or the spiders the “unknown” part? Besides that, the authors presented data of nine species of spiders, but only the data of 4 are absent in literature.”
Reply:
The “unknown” is related to the silk of those 4 species that are absent in literature. Thus, we have substituted the title with “The mechanical properties and Weibull parameters of 9 spider silks (4 of them unknown in literature) and related scaling laws on the fracture strength”.
3) “Page 2, line 80: “Fera” should be changed to “fera”
Reply:
We modified the manuscript accordingly, see marked copy.
4) “Pag 2, line 88: “second” should be changed to “s””
Reply:
We modified the manuscript accordingly, see marked copy.
5) “Several text spaces are missing or wrong along all the manuscript. For example, pag 2, line 88: “1% /second” or “0.1mm/s”; or pag 4, line 142: “T>F”. Please revise this along all the manuscript.”
Reply:
We modified the manuscript accordingly, see marked copy.
6) “Page 4, line 158. Remove the sentence “The mechanical properties of the silks of Ancylometes bogotensis, Ceratogyrus marshalli, Linothele fallax, and Phoneutria fera were not reported in literature.” from the caption of Table 1. There is no need to have this information in the caption. Beside that, its already in the line 148-149.”
Reply:
We modified the manuscript accordingly, see marked copy.
7) “All the graphics should be revised. Minor and major ticks in the xx and yy axis’s are missing and the graphics should be redesigned to be visually more attractive and easy to analyze data.”
Reply:
We modified the manuscript accordingly, see marked copy.
8) “Pag. 8, line 223: Is there any substantial reason to have “Bioinspired” and “Bio-compatible” with the first letter capitalized? “Bio-compatible” should be changed to “biocompatible”.”
Reply:
We modified the manuscript accordingly, see marked copy.
9) “Pag. 8, line 223 “In this work we have studied the silk of nine species (some of them unknown in literature) by measuring the stress-strain curves and obtaining their mechanical properties strength of their silk.”. It must be rewritten or “…of their silk” eliminated.”
Reply:
We modified the manuscript accordingly, see marked copy.
10) “Page 8, line 230: “The former presented low Weibull modulus (between 1 and 2) maintaining constant in time fracture probability and thus mechanical behavior7,29,30 (comparable to what can be found in literature on this types of silks7).” This sentence is confused, it must be rewritten. “...this types…” should be “…these types…”.”
Reply:
We modified the manuscript accordingly, see marked copy.
11) “Are any of the presented data surprising for the authors?
How and why were these spider species chosen for this study?
The presented data (not only the mechanical behavior) should be compared with the ones of the species that are already available in literature.”
Reply:
The data that we obtained were not surprising in relation to the initial guess.
In our lab we have many species of spiders. Here the species analysed were nine: 4 unknowns in the literature and 5 already known in the literature, these are Araneus diadematus, Grammostola rosea, Nuctenea umbratica, Cupiennius salei and Zygiella x-notata and were chosen to have a comparison with the literature. We have thus added:
Lines 84-86:
“These species were selected because of the availability of adult females in our farm. Cupiennius salei was chosen because it is a massive wandering spider that produces easily collectable dragline.”
Lines 164-165:
“The other species analysed in this work presented mechanical properties comparable with those in literature9–11.”
Added references:
- Yazawa, K., Malay, A. D., Masunaga, H., Norma-Rashid, Y. & Numata, K. Simultaneous effect of strain rate and humidity on the structure and mechanical behavior of spider silk. Commun. Mater. 1–10 (2020). doi:10.1038/s43246-020-0011-8
- Pérez-Rigueiro, J., Elices, M., Llorca, J. & Viney, C. Tensile properties of Argiope trifasciata drag line silk obtained from the spider’s web. J. Appl. Polym. Sci. 82, 2245–2251 (2001).
- Rinne, H. The Weibull Distribution. A handbook. (2009).
- Prez-Rigueiro, J., Elices, M. & Llorca, C. V. Tensile properties of Argiope trifasciata drag line silk obtained from the spider’s web. J. Appl. Polym. Sci. 82, 2245–2251 (2001).
- Peterlik, H. The validity of Weibull estimators. J. Mater. Sci. 30, 1972–1976 (1995).
- Denny, M. W. the Physical Properties of Spider’S Silk and Their Role in the Design of Orb-Webs. J. Exp. Biol. 65, 483–506 (1976).
- Ortlepp, C. & Gosline, J. M. The scaling of safety factor in spider draglines. J. Exp. Biol. 211, 2832–2840 (2008).
- Greco, G., Wolff, J. & Pugno, N. M. Strong and tough silk for resilient attachment discs: the mechanical properties of piriform silk, in the spider Cupiennius salei (Keyserling, 1877). Front. Mater. In press., (2020).
- Swanson, B. O., Blackledge, T. A., Beltrán, J. & Hayashi, C. Y. Variation in the material properties of spider dragline silk across species. Appl. Phys. A Mater. Sci. Process. 82, 213–218 (2006).
- Swanson, B. O., Anderson, S. P., DiGiovine, C., Ross, R. N. & Dorsey, J. P. The evolution of complex biomaterial performance: The case of spider silk. Integr. Comp. Biol. 49, 21–31 (2009).
- Agnarsson, I., Kuntner, M. & Blackledge, T. A. Bioprospecting finds the toughest biological material: Extraordinary silk from a giant riverine orb spider. PLoS One 5, 1–8 (2010).
Round 2
Reviewer 1 Report
My greatest problem with the original manuscript was the statistical analysis and the conclusions drawn. I have considered the authors' responses, I have thought more about this problem, and I have done some additional reading. Yet, I am now more convinced than before that the statistical analysis of this manuscript -- and especially the conclusions drawn from this analysis -- are seriously flawed, and the manuscript must not be published before these issues are corrected.
The problem really arises from the author's interpretation of the outcome of their null hypothesis significance testing (NHST). Especially the correct interpretation of the outcome of this technique (p-values) is highly complicated and has led to huge problems in science, in general. This is why the use of p-values has come under extreme scrutiny in the past years, to the extent that some journals have banned them. This study exemplifies such problems.
The null hypothesis the authors have used is that the data of the five different samples (4 mechanical parameters calculated from stress-strain curves taken at 5 different strain rates) come from an identical population. The results of the corresponding analysis are quite small p-values of 0.0461, 0.000044, 5.4*10^(-19), and 0.0016 for strain, strength, stiffness, and toughness, respectively. These values are generally considered to be all in the "significant" range (often defined as p<0.05 or p<0.01, etc.). However, all that we can conclude from this is that the observed data is unlikely, if the null hypothesis were correct. In the best case, this means that we can reject the null hypothesis (if none of the underlying assumptions were violated). In other words, all we can say is that the observed results obtained at 5 different strain rates SOMEHOW deviate from what the data would look like if there was no effect of the strain rate whatsoever. In no way this analysis tells us HOW the data deviates from what would be expected assuming the null hypothesis.
In L.173-174 of their revised manuscript, the authors state the following: "Similarly, for the Young’s modulus, we observed a significant increase (p-value=5.4*10-19 ) with strain rate (see trend line in Figure 1c)." But unfortunately, this statement is not supported by the presented statistical analysis whatsoever. Therefore, this is a completely unsupported claim. From the rejection of the null hypothesis (that the strain rate has no effect on the modulus), one can in no way conclude that the modulus increases a a function of the strain rate. As a matter of fact, rejection of the null hypothesis could as well mean the opposite: that the modulus is decreasing as a function of strain rate. However, there are many other ways, in which the data could have varied to cause a rejection of the null hypothesis, ways that would neither constitute and increase or a decrease of the Young's modulus as a function of strain rate. Adding a "trend line" does in no way support the argument of the authors. To make such an analysis meaningful, one would have to provide statistical support for the conclusions drawn. In other words, a statistical analysis would have to support the authors' claim -- that the modulus increases as a function of the strain rate. However, the authors present no such support of their claim. For instance, one could add confidence intervals to the slope fitted to the data. However, looking at how large the variances of the 5 distributions are, I would not be surprised if such confidence intervals extend to zero or even negative slopes -- which would mean that the authors' claims can simply not be supported on the basis of the data.
The authors made the same mistake in their interpretation of the other three mechanical quantities (strain at break, strength, and toughness). About the strength the state the following in L.170-172 of their revised manuscript: "For the strength we observed a significant difference (p-value=0.000044) among all the strain rates. In particular, the strength slightly increased (see trend line in Figure 1b) with the strain rate." To state "In particular,..." in light of the fact that their statistical analysis does simply not consider whether the strength increases or decreases is highly misleading.
The claim about strain at break is similarly unsupported (L.166-169): "The strain at break, displayed in Figure 1a, resulted to be significantly different among all the groups (p-value=0.0461). In particular, it slightly decreased with the increase of the strain rate. This was stated by looking at the trend line (Figure 1a)." Yes, the statistical analysis shows that the strain at break somehow varies, but in no way supports a decrease as a function of strain rate.
Same for the toughness (L.176-177): "The toughness modulus follows the tendency of the strength and Young’s modulus (see trend line in Figure 1d), but with a more moderate increase (p-value=0.0016)." Again, the statistical analysis in no way supports the claimed trend. In line 179-181 they continue: "In this case, however, it is important to notice the huge scattering of the data. Moreover, we highlight the fact that the highly different values at 0.15 mm/s have particularly affected the slope of the trend line." Unfortunately, the admission that some of their data points show significant scatter cannot make up for the fact that the authors drew wrong conclusions from a statistical analysis.
There have been many discussions around the overuse and misuse of p-values all over the literature, including in the most famous journals, such as Nature. One of the commonly suggested "cures" of the p-value problem was to talk more about effect size. And this is exactly where the authors fall short. They present no rigorous statistical analysis of the effect size of their claim. They present statistical support for the rejection of the null hypothesis, but they do not present any statistical support of their actual hypothesis whatsoever (for instance, "Young's modulus increases with increasing strain rate"). For a manuscript that attempts to address the "absence of deep statistical investigation on the mechanical properties of spider silks" (see the abstract), this is not satisfying.
Author Response
- “My greatest problem with the original manuscript was the statistical analysis and the conclusions drawn. I have considered the authors' responses, I have thought more about this problem, and I have done some additional reading. Yet, I am now more convinced than before that the statistical analysis of this manuscript -- and especially the conclusions drawn from this analysis -- are seriously flawed, and the manuscript must not be published before these issues are corrected.”
Reply:
We thank the reviewer for insisting on this p-value critical point. We agree that some classical paradoxes are associated with this type of analysis. When we referred to the statistical analysis of spider silk we referred mainly to Weibull statistics. We have thus minimized (by referring to pairwise comparison) the other (p-value) statistical analysis in order to avoid misunderstanding and have added the “effect-size” analysis suggested by the referee. We hope that the current version will fully satisfy the final concerns of the referee. See point by point reply below.
- “The problem really arises from the author's interpretation of the outcome of their null hypothesis significance testing (NHST). Especially the correct interpretation of the outcome of this technique (p-values) is highly complicated and has led to huge problems in science, in general. This is why the use of p-values has come under extreme scrutiny in the past years, to the extent that some journals have banned them. This study exemplifies such problems. The null hypothesis the authors have used is that the data of the five different samples (4 mechanical parameters calculated from stress-strain curves taken at 5 different strain rates) come from an identical population. The results of the corresponding analysis are quite small p-values of 0.0461, 0.000044, 5.4*10^(-19), and 0.0016 for strain, strength, stiffness, and toughness, respectively. These values are generally considered to be all in the "significant" range (often defined as p<0.05 or p<0.01, etc.). However, all that we can conclude from this is that the observed data is unlikely, if the null hypothesis were correct. In the best case, this means that we can reject the null hypothesis (if none of the underlying assumptions were violated). In other words, all we can say is that the observed results obtained at 5 different strain rates SOMEHOW deviate from what the data would look like if there was no effect of the strain rate whatsoever. In no way this analysis tells us HOW the data deviates from what would be expected assuming the null hypothesis.”
Reply:
We thank the reviewer for this comment. In order to simplify the interpretation of the data and to remove the misinterpretation of the p-value meaning, we now have considered the p-values only in pairwise comparison (comparison between two groups), which is the more usual procedure to compare different group.
Consequently, we have added/modified the following lines:
Lines 133-134
“One-way Analysis of Variance (pairwise comparison) was performed to compare the mechanical properties of the fibres obtained under different parameters.”
Moreover, we added in S.I. the tables were all this p-values (and Cohen dC for effect size) are depicted.
- “In L.173-174 of their revised manuscript, the authors state the following: "Similarly, for the Young’s modulus, we observed a significant increase (p-value=5.4*10-19 ) with strain rate (see trend line in Figure 1c)." But unfortunately, this statement is not supported by the presented statistical analysis whatsoever. Therefore, this is a completely unsupported claim. From the rejection of the null hypothesis (that the strain rate has no effect on the modulus), one can in no way conclude that the modulus increases a a function of the strain rate. As a matter of fact, rejection of the null hypothesis could as well mean the opposite: that the modulus is decreasing as a function of strain rate. However, there are many other ways, in which the data could have varied to cause a rejection of the null hypothesis, ways that would neither constitute and increase or a decrease of the Young's modulus as a function of strain rate. Adding a "trend line" does in no way support the argument of the authors. To make such an analysis meaningful, one would have to provide statistical support for the conclusions drawn. In other words, a statistical analysis would have to support the authors' claim -- that the modulus increases as a function of the strain rate. However, the authors present no such support of their claim. For instance, one could add confidence intervals to the slope fitted to the data. However, looking at how large the variances of the 5 distributions are, I would not be surprised if such confidence intervals extend to zero or even negative slopes -- which would mean that the authors' claims can simply not be supported on the basis of the data. The authors made the same mistake in their interpretation of the other three mechanical quantities (strain at break, strength, and toughness). About the strength the state the following in L.170-172 of their revised manuscript: "For the strength we observed a significant difference (p-value=0.000044) among all the strain rates. In particular, the strength slightly increased (see trend line in Figure 1b) with the strain rate." To state "In particular,..." in light of the fact that their statistical analysis does simply not consider whether the strength increases or decreases is highly misleading.The claim about strain at break is similarly unsupported (L.166-169): "The strain at break, displayed in Figure 1a, resulted to be significantly different among all the groups (p-value=0.0461). In particular, it slightly decreased with the increase of the strain rate. This was stated by looking at the trend line (Figure 1a)." Yes, the statistical analysis shows that the strain at break somehow varies, but in no way supports a decrease as a function of strain rate.Same for the toughness (L.176-177): "The toughness modulus follows the tendency of the strength and Young’s modulus (see trend line in Figure 1d), but with a more moderate increase (p-value=0.0016)." Again, the statistical analysis in no way supports the claimed trend. In line 179-181 they continue: "In this case, however, it is important to notice the huge scattering of the data. Moreover, we highlight the fact that the highly different values at 0.15 mm/s have particularly affected the slope of the trend line." Unfortunately, the admission that some of their data points show significant scatter cannot make up for the fact that the authors drew wrong conclusions from a statistical analysis.”
Reply:
We thank the reviewer for the comment. For the sake of clarity and correctness, we removed the assertions on the trendline, and we limited our statements on the basis of the pairwise p-values comparison.
- “There have been many discussions around the overuse and misuse of p-values all over the literature, including in the most famous journals, such as Nature. One of the commonly suggested "cures" of the p-value problem was to talk more about effect size. And this is exactly where the authors fall short. They present no rigorous statistical analysis of the effect size of their claim. They present statistical support for the rejection of the null hypothesis, but they do not present any statistical support of their actual hypothesis whatsoever (for instance, "Young's modulus increases with increasing strain rate").”
Reply:
We thank the reviewer for suggesting the effect size analysis. We added such analysis by considering the parameter introduced by Cohen (dC).
We added/modified the following lines
Lines 154-170:
“2.5. Effect size
The p-value is affected by the sample’s dimension. Larger samples are likely to give statistically significant differences even though such differences are small42. For this reason, it is important to consider the size effect, which is a measure of the magnitude of such difference43. We based our analysis on the parameters introduced by Cohen43. By assuming that the two compared population have the same variance, a pooled standard deviation can be defined as
(see uploaded manuscript)
where n1 and n2 are the dimension of the two groups compared, and s1 and s2 their standard deviation. With s, Cohen defined the following parameter
(see uploaded manuscript)
where m1 and m2 are respectively the means of the two groups. Based on Cohen43 and Sawiloski44, dc helps to define qualitatively the magnitude of the difference of the means as very small (dc ≥ 0.01, i.e. circa 100% distributions’ overlap), small (dc ≥ 0.20, i.e. circa 85% distributions’ overlap), medium (dc ≥ 0.50, i.e. circa 67% distributions’ overlap), large (dc ≥ 0.8, i.e. circa 53% distributions’ overlap), very large (dc ≥ 1.20, i.e. circa 40% distributions’ overlap), and huge (dc ≥ 2.0, i.e. circa 19% distributions’ overlap).”
Lines 188-191:
“The strain at break, displayed in Figure 1a (p-values and dc coefficient in Table S1), resulted to be significantly higher at 0.11 mm/s with respect to the 0.08 mm/s, 0.10 mm/s, and 0.17 mm/s strain rates (with a medium size effect).”
Lines 194-196:
“For the strength, displayed in Figure 1b (p-values and dc coefficient in Table S2) we observed significant higher values for the strain rates 0.15 mm/s with respect all the others (with a large size effect).”
Lines 200-207:
“Similarly, for the Young’s modulus (Figure 1c, p-values and dc coefficient in Table S3), we observed a significant higher value for the strain rates of 0.10 mm/s with respect to the 0.08 mm/s (with a small size effect). 0.15 mm/s strain rates presented the significantly highest Young’s modulus with respect to 0.08 mm/s (very large size effect), 0.10 mm/s (medium size effect), 0.11 mm/s (very large size effect), and 0.17 mm/s (medium size effect). 0.11 mm/s strain rates gave lower values with respect to 0.08 mm/s (with a large size effect), 0.10 mm/s (with a large size effect), and 0.17 mm/s (with a large size effect). Finally, the strain rates of 0.17 mm/s gave lower Young’s modulus values with respect to the strain rates of 0.10 mm/s (with a very small size effect).”
Lines 210-211:
“The toughness modulus (Figure 1d, p-values and dc coefficient in Table S4) measured at 0.15 mm/s was significantly higher with respect all the others (with a medium size effect).”
Lines 230-236:
“On the other hand, in Figure 2a and Table S5 it is possible to notice a slightly decrease of the diameter for longer samples. In particular, the diameter’s value of the 0.55 cm length was the higher than the one at 0.75 cm (with a medium size effect), and higher than the ones at 1.0 cm, 1.25 cm, and 1.5 cm (with a large size effect). Moreover, the diameter’s value of the samples 0.75 cm long was higher than that of 1.0 cm (with a medium size effect). On the other hand, although the difference between the diameters of the samples at 0.75 cm and those at 1.25 cm and 1.5 cm were significative, their size effects were small.”
Lines 246-252:
“The strength of the silk at different lengths is reported in Figure 2b (for the p-values and the Cohen’s coefficient see Table S6). Respectively, the values measured were 931 ± 345 MPa, 805 ± 371 MPa, 754 ± 315 MPa, 789 ± 317 MPa, and 515 ± 260 MPa. We observed, thus, a slightly significant decrease in strength in longer fibres. In particular, the fibres 0.55 cm long were significantly stronger with respect those 1.0 cm long (with a medium size effect) and those 1.5 cm long (with a very large size effect). Moreover, the fibres 1.5 cm long were significantly weaker than those 0.75 cm, 1.0 cm, and 1.25 cm long (with a large size effect).”
- “For a manuscript that attempts to address the "absence of deep statistical investigation on the mechanical properties of spider silks" (see the abstract), this is not satisfying.”
Reply:
We removed the word “deep” and we hope that now the manuscript can be considered for publication.
Added references:
- Sullivan, G. M. & Richard Feinn. Using Effect Size—or Why the P Value Is Not Enough. J. Grad. Med. Educ. Editorial, 279–282 (2012).
- Cohen, J. Statistical power analysis for the behavioral sciences. (Lawrence Erlbaum Associates, 1988).
- Sawilowsky, S. S. New Effect Size Rules of Thumb. J. Mod. Appl. Stat. Methods 8, 597–599 (2009).
